# Hydrogen and Deuterium Incorporation in ZnO Films Grown by Atomic Layer Deposition

Sami Kinnunen[1,2,*], Manu Lahtinen[3], Kai Arstila[1,2] and Timo Sajavaara[1,2]

1    Department of Physics, University of Jyvaskyla, P.O. Box 35, FI-40014 Jyväskylä, Finland;
     kai.arstila@jyu.fi (K.A.); timo.sajavaara@jyu.fi (T.S.)
2    Nanoscience Center, University of Jyvaskyla, P.O. Box 35, FI-40014 Jyväskylä, Finland
3    Department of Chemistry, University of Jyvaskyla, P.O. Box 35, FI-40014 Jyväskylä, Finland;
     manu.k.lahtinen@jyu.fi
*    Correspondence: samantki@jyu.fi

**Abstract:** Zinc oxide (ZnO) thin films were grown by atomic layer deposition using diethylzinc (DEZ) and water. In addition to depositions with normal water, heavy water ($^2H_2O$) was used in order to study the reaction mechanisms and the hydrogen incorporation at different deposition temperatures from 30 to 200 °C. The total hydrogen concentration in the films was found to increase as the deposition temperature decreased. When the deposition temperature decreased close to room temperature, the main source of impurity in hydrogen changed from $^1H$ to $^2H$. A sufficiently long purging time changed the main hydrogen isotope incorporated in the film back to $^1H$. A multiple short pulse scheme was used to study the transient steric hindrance. In addition, the effect of the storage of the samples in ambient conditions was studied. During the storage, the deuterium concentration decreased while the hydrogen concentration increased an equal amount, indicating that there was an isotope exchange reaction with ambient $H_2$ and/or $H_2O$.

**Keywords:** ZnO; ALD; heavy water; diethylzinc; ToF-ERDA





## 1. Introduction

Atomic layer deposition (ALD) is a thin film deposition technique that is widely adopted in integrated circuits [1] and other modern devices such as in organic light emitting diodes (OLEDs) [2] and perovskite solar cells [3]. Unique to ALD is its ability to control the film thickness with subnanometer precision and to produce a conformal film on top of complex three-dimensional objects as well as on porous substrates [4,5].

Zinc oxide, ZnO, is an optically transparent wide band gap semiconductor ($E_g \sim 3.3$ eV) [6]. The electrical properties of ZnO can be tailored with doping, which makes it a versatile material for numerous applications [7]. The unique assets of ALD have attracted the attention of ALD-grown ZnO films for various applications. For example, the excellent conformality enables the deposition of ZnO on high surface area powders used in catalysis [8]. Doped ZnO can be used as transparent conductive oxide (TCO) for optoelectronics [9] and ZnO can be deposited at near room temperature [6,10–13], making ALD a suitable deposition method for temperature-sensitive substrates such as polymers. Recently, spatial atomic layer deposition (SALD) has been harnessed for high throughput and large area depositions, making ALD more interesting for industrial use [6]. In addition, ALD ZnO has been studied for its reversible wettability [14,15], and it has been found to have antibacterial properties [16].

The atomic layer deposition of ZnO using diethylzinc (DEZ) and water as precursors has been widely studied. ZnO films deposited with DEZ and water are polycrystalline with a hexagonal wurtzite crystal structure [10,17]. While the crystal structure itself does not change, the preferred orientation of the crystals can change with the deposition temperature [10,13,18]. In addition, the substrate [17] and the process parameters such as pulse and purging times [19] have been found to affect the crystal orientation.

The ideal reaction mechanism for DEZ and $H_2O$ is as follows [20]:

$$|| - OH + Zn(CH_2CH_3)_2\,(g) \longrightarrow || - O - Zn(CH_2CH_3) + CH_3CH_3\,(g) \qquad (1)$$

$$|| - OH - Zn(CH_2CH_3) + H_2O\,(g) \longrightarrow || - O - Zn - OH + CH_3CH_3\,(g). \qquad (2)$$

In the first half reaction (1), DEZ reacts with the surface OH-group, releasing ethane. The half reaction stops when all the OH-groups have reacted and the whole surface is saturated with monoethylzinc. In the second half reaction (2), water is pulsed and removes ethyl groups from zinc atoms and renews the surface with OH-groups. In the ideal case, the saturation of the surface is reached when all of the hydroxyl or ethyl groups are removed from the surface.

However, in reality, the reaction is more complex, and less than a monolayer of material is deposited in each cycle. This phenomenon is often related to the steric hindrance caused by the ligand molecules. Bulky ligands, such as ethyl groups, can cover reactive sites, and the saturation of growth is reached before every reactive site is occupied [21]. At high temperatures, the desorption of molecules providing reactive sites can also decrease the growth-per-cycle (GPC), and the self-limiting reaction is terminated due to the lack of reactive sites rather than steric hindrance [21].

Furthermore, at low temperatures, the saturation of the surface can be due to the low reactivity of the ligands. The termination of the half cycle for the low-temperature deposition of $Al_2O_3$ from trimethylaluminium (TMA) and $H_2O$ was experimentally studied by Vandalon and Kessels [22]. They showed that after the $H_2O$ pulse, there were persistent $CH_3$-groups at the surface. Their conclusion was that, at low temperatures, water is not reactive enough towards the methyl surface species, resulting in saturation. Similar results were obtained by Mackus et al. for DEZ and $H_2O$ [23]. There is also strong theoretical evidence of persistent surface groups. Both the DEZ and $H_2O$ half cycles were studied computationally by Weckman and Laasonen [24,25]. They found out that DEZ is reactive towards water even at room temperature, but water is not able to remove all the ethyl groups and the water pulse would therefore be the growth-limiting step in the reaction.

Some sensitive substrates, such as organic materials and plastics, may require a low deposition temperature. Reaction rates at these temperatures are not only slower but also the mechanism of saturation of the half cycle might be different than at higher temperatures. As a result, significant amounts of precursor-derived impurities, such as carbon and hydrogen, are incorporated in the film due to incomplete reactions. While hydrogen incorporation is sometimes even desired [26], more often, high-purity films are required, and hydrogen impurities can have a negative effect on the film properties [27]. The hydrogen concentration of ALD ZnO films has also been found to be related to the conductivity of the films [28,29].

In this study, we used heavy water, $^2H_2O$, as an oxygen source in order to probe the impurities and to study the reaction mechanisms of ZnO films deposited at low temperatures using DEZ. Hydrogen originating from the precursors is a common impurity in ALD films, and with $^2H_2O$, it is possible to distinguish whether the hydrogen in the film originates from DEZ or water. In addition, this can reveal valuable information regarding the reaction mechanisms. Heavy water is, in principle, chemically identical to normal water as none of the elements change. However, the heavier hydrogen isotope has some effect on the reactions, as discussed in the text below.

Earlier reaction mechanism studies have been performed with precursors containing rare stable isotopes, such as heavy water [30–36]. There are a few techniques that can be utilized to differentiate isotopes from each other. Quadrupole mass spectrometry (QMS) can be used to detect reaction side-products in the gas phase [31–33]. Ion beam techniques, such as time-of-flight elastic recoil detection analysis (ToF-ERDA) [30,37] and secondary ion beam mass spectrometry (SIMS) [35], can detect different isotopes within the film. In addition, IR spectroscopy can be used to resolve different isotopes [34]. In our study,

we utilized ToF-ERDA, which can directly and quantitatively measure hydrogen and its isotopes as well as the elemental composition of the films.

## 2. Materials and Methods

Thin films were deposited using a Beneq TFS 200 ALD-reactor. Nitrogen from an Inmatec PN 1150 nitrogen generator (99.999% purity) was used as a purging gas. Diethylzinc (Strem Chemicals, min. 95%) was used as a zinc precursor. Both normal deionized water $^1H_2O$ and heavy water $^2H_2O$ (Medical Isotopes Inc., Pelham, NH, USA. 99.9%) were used as oxygen sources. Depositions were performed at 1–2 mbar of base pressure with a constant (300 sccm) $N_2$ flow. The effect of the temperature on ZnO films was studied using 150 ms and 500 ms pulses for DEZ and $^1H_2O/^2H_2O$, respectively. The purging time after the DEZ pulse was 10 s, and 20 s after $^1H_2O/^2H_2O$. In addition, ZnO samples were deposited using variety of purging times at 60 °C and with a multiple short pulsing (MSP) scheme at 40 °C.

Atomic force microscopy (AFM), helium ion microscopy (HIM) and powder X-ray diffraction (XRD) were used to study the crystallinity of the deposited films. AFM imaging was done using Bruker Dimension Icon in the peak force tapping mode. XRD measurements were carried out using a Malvern Pananalytical X'Pert PRO diffractometer with Cu $K_\alpha$ radiation ($\lambda$ = 1.54187 Å via Ni $\beta$-filter; 45 kV, 40 mA). Data processing and search-match phase analyses were carried out using the program X'pert HighScore Plus v. 4.9 and ICDD-PDF4+ database (version 2020) [38,39]. Cross sections of cleaved ZnO samples were imaged with a Carl Zeiss Orion NanoFab helium ion microscope using a 30 keV helium beam at a 45° angle.

Thin film elemental depth profiles were measured with ToF-ERDA using a 13.615 MeV $^{127}I^{7+}$ ion beam [40]. Analysis was done using Potku-software [41].

Film thicknesses were measured with a Rudolph AUTO EL III ellipsometer using a 632.8 nm wavelength.

## 3. Results and Discussion

All the deposited ZnO films were found to be slightly oxygen rich according to the ToF-ERDA measurements. The measured O/Zn ratio varied between 1.33 and 1.09, and the O/Zn ratio decreased with increasing deposition temperature. Due to the scattering effects, the amount of Zn in the film was slightly underestimated [42]. However, oxygen rich ZnO films deposited with ALD, especially at low deposition temperatures, have been reported previously [10,27,43–45]. The high oxygen concentration in the ZnO films was attributed to the zinc vacancies [44] in the crystal as well as to the oxygen interstitials [43]. In addition, hydrogen impurities were proposed to occupy the Zn-sites in the crystal [27].

The growth-per-cycle increased with increasing deposition temperature, reaching 1.7 Å/cycle at 150 °C when $^1H_2O$ was used (Figure 1a and Table 1). When the deposition temperature was further increased to 200 °C, the GPC decreased. This expected and well-documented decrease in GPC has been attributed to both the loss of OH-groups at elevated temperatures and the desorption of precursors before reacting on the surface [11,20,25]. The loss of OH-groups decreases the number of reactive sites for DEZ and results in a lower GPC. The GPC results in this study are somewhat lower than previously reported in the literature [10,11,20] but still comparable. The observed differences could have originated from multiple sources such as reactor design, reactor pressure and purging times [46]. When heavy water was used instead of normal water (Figure 1a), the GPC was significantly lower, especially at low temperatures. This can be attributed to the kinetic isotope effect. The use of a heavier hydrogen isotope increased the activation energy of the reaction, which slowed down the reaction rate. Therefore, the kinetic isotope effect can be thought of as equivalent to the decrease in the temperature. The effect became less significant as the temperature increased and there was enough energy for the reaction to reach completion. The kinetic isotope effect is utilized, for example, in chemical reaction

mechanism studies [47], where an isotopic substitution changes the reaction rate if a bond at or near the substitute plays a role in the reaction.

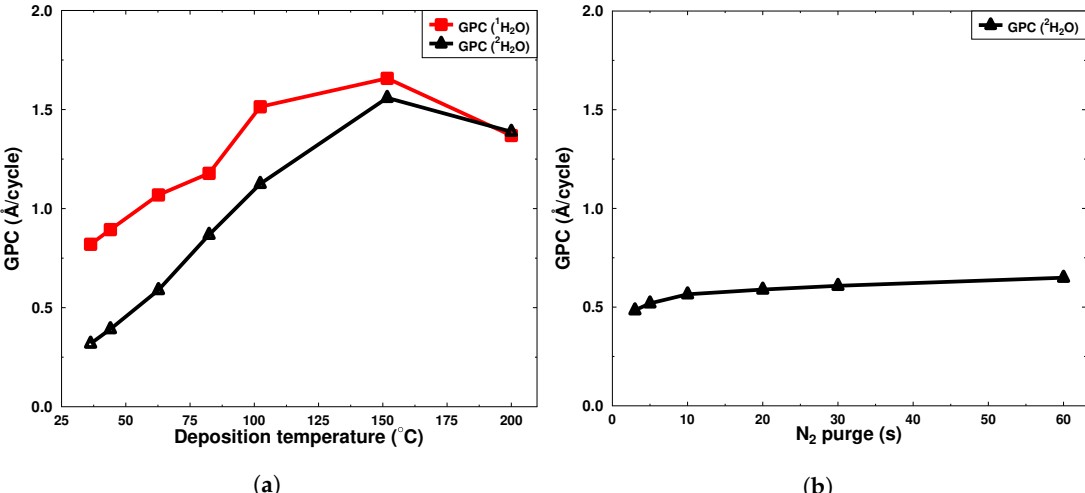

(**a**)　　　　　　　　　　　　　　　　　　　　　　　　　(**b**)

**Figure 1.** (**a**) Growth-per-cycle as a function of the deposition temperature. One ALD cycle consisted of a 150 ms DEZ pulse, 10 s purge, 500 ms $^1H_2O$/$^2H_2O$ pulse and 20 s purge. (**b**) Growth-per-cycle as a function of the purging time using DEZ and $^2H_2O$ as precursors at 60 °C. The purging times after both precursors were kept the same when the purging time was 3, 5 or 10 s. For the longer purging times, only the purge after $^2H_2O$ was increased, and the purge after DEZ was kept at 10 s.

**Table 1.** The main results of ZnO samples deposited with both $^1H_2O$ and $^2H_2O$ at different deposition temperatures. Each cycle consisted of a 150 ms DEZ pulse, 10 s purge, 500 ms $^1H_2O$/$^2H_2O$ pulse and 20 s purge.

| Sample | T (°C) | $^1H$ (at.%) | $^2H$ (at.%) | C (at.%) | Thickness (nm) | RMS Roughness (nm) | GPC (Å/cycle) |
|---|---|---|---|---|---|---|---|
| DEZ+$^2H_2O$ | | | | | | | |
| D30 | 30 | 10.7 | 0.7 | 1.3 | 45 | 3.6 | 0.32 |
| D40 | 40 | 8.7 | 1.0 | 0.8 | 47 | 3.5 | 0.39 |
| D60 | 60 | 1.4 | 3.3 | 0.3 | 59 | 3.4 | 0.59 |
| D80 | 80 | 0.7 | 2.4 | 0.2 | 87 | 5.7 | 0.87 |
| D100 | 100 | 0.5 | 1.6 | 0.2 | 113 | 7.5 | 1.13 |
| D150 | 150 | 0.1 | 0.7 | 0.2 | 156 | 6.2 | 1.56 |
| D200 | 200 | >0.1 | 0.4 | >0.1 | 139 | - | 1.39 |
| DEZ+$^1H_2O$ | | | | | | | |
| H30 | 30 | 7.8 | - | 0.5 | 82 | 4.5 | 0.82 |
| H40 | 40 | 5.3 | - | 0.3 | 89 | 4.3 | 0.89 |
| H60 | 60 | 3.2 | - | 0.1 | 107 | 4.6 | 1.07 |
| H80 | 80 | 2.1 | - | 0.2 | 118 | 6.8 | 1.18 |
| H100 | 100 | 1.6 | - | 0.2 | 155 | 9.2 | 1.55 |
| H150 | 150 | 1.0 | - | 0.2 | 133 | 3.1 | 1.66 |
| H200 | 200 | 0.4 | - | 0.1 | 137 | - | 1.37 |

The increase of the purging time slightly increased the growth per cycle (Figure 1b and Table 2), but the effect was only minor at 60 °C when DEZ and $^2H_2O$ were used. Park et al. reported that long purging times have a considerable impact on the film properties [48]. In their study, they observed that at 170 °C, the GPC decreased from 2.2 Å/cycle to 1.6 Å/cycle when the purging time after the $^1H_2O$ pulse was increased from 20 s to 120 s, thus leading them to conclude that the decrease in the GPC was due to the loss of OH-groups via dehydration, which led to a decreased concentration of the reaction sites for DEZ molecules in the next precursor pulse. A comparison with our results indicates that the effect of purging on GPC is dependent on the deposition temperature. The GPC at 60 °C is not

limited due to the loss of OH-groups during the purge step but rather due to the slow reactivity of the precursors. Longer purging times at low temperatures are often favored in order to avoid uncontrolled CVD-like growth resulting in a high GPC. However, no indication of CVD growth was detected, even at very short purging times, as seen in Figure 1b. The increase of the GPC with increasing purging time at 60 °C could have been due to slow reactions and the desorption of by-products. If the by-products were not desorbed from the surface when the next precursor arrived, they could block reactive sites which in turn would reduce the GPC. This transient steric hindrance is discussed later in the text. At higher deposition temperatures, the reactions and desorption of by-products would likely be fast enough so that other factors, such as the concentration of the reactive sites, would become the limiting factors to the GPC.

**Table 2.** The main results of the ZnO samples deposited at 60 °C with varying purging times. The precursor pulse times were kept at 150 ms and 500 ms for DEZ and $^2H_2O$, respectively.

| Sample | $N_2$ Purge (s) | $^1H$ (at.%) | $^2H$ (at.%) | C (at.%) | Thickness (nm) | RMS Roughness (nm) | GPC (Å/cycle) |
|---|---|---|---|---|---|---|---|
| N3 | 3 | 8.5 | 1.0 | 0.7 | 48 | 3.3 | 0.48 |
| N5 | 5 | 3.4 | 4.2 | 0.7 | 52 | 2.7 | 0.52 |
| N10 | 10 | 1.9 | 3.4 | 0.3 | 57 | 3.3 | 0.57 |
| N20 | 20 | 1.4 | 3.3 | 0.3 | 59 | 3.4 | 0.59 |
| N30 | 30 | 1.3 | 3.1 | 0.3 | 61 | 3.1 | 0.61 |
| N60 | 60 | 1.0 | 2.7 | 0.2 | 65 | 3.4 | 0.65 |

The effect of using $^2H_2O$ instead of $^1H_2O$ on film crystallinity and surface morphology was studied with XRD, AFM and HIM. The diffraction peaks characteristic to the wurtzite structure in Figure 2 (JCPDS 01-084-6784 [49]) show a transformation of preferred orientation from (002) to (100), as reported by Malm et al. [10] and also by Cai et al. [11] and Guziewicz et al. [12]. At low temperatures (40 °C and 60 °C), the preferred orientation of the crystals was (002), and this transformed to (100) at higher temperatures (100 °C). However, the crystallinity in the films was quite low, as indicated by the broad peak profiles and low intensities of the characteristic peaks. The films deposited with $^1H_2O$ were more crystalline than the films deposited with $^2H_2O$. Some of this difference can be attributed to thinner films due to the lower GPC, as seen in Table 1. The film deposited at 40 °C with $^2H_2O$ was 47 nm thick, while the film deposited with $^1H_2O$ was 89 nm thick. At 100 °C, the thicknesses were 113 and 155 nm, respectively. However, the trend of change in the orientation was similar for both $^1H_2O$ and $^2H_2O$.

The mtomic force microscope micrographs in Figure 3 confirm the transformation of the preferred crystal orientation from (002) to (100). The change in crystal orientation is visible in AFM micrographs at lower temperatures than in XRD patterns. The same transformation is also evident from the HIM images in Figure 4a,b. Cross-sectional HIM images also reveal that neither the crystallinity nor the orientation changed considerably within the film. HIM images of ZnO films deposited at 40, 60, 100 and 150 °C with both normal and heavy water are presented in Supplementary Figures S1 and S2. The root-mean-square (RMS) roughness (calculated from the AFM micrographs) of the films with a preferred (002) orientation varied from 3.4 to 4.5 nm, as seen in Table 1. The roughness increased at higher temperatures when the (100) orientation became visible, with its highest value of 9.2 nm in the case of the sample deposited with $^1H_2O$ at 100 °C. When $^1H_2O$ was used as an oxygen source, crystals with (100) orientation were visible already at the deposition temperature of 60 °C. In the case of heavy water, the (100) oriented crystals became visible only at 80 °C and above as seen in Figure 3. This also supports the earlier claim that the use of $^2H_2O$ has a similar effect on the deposition process as the decrease in the deposition temperature.

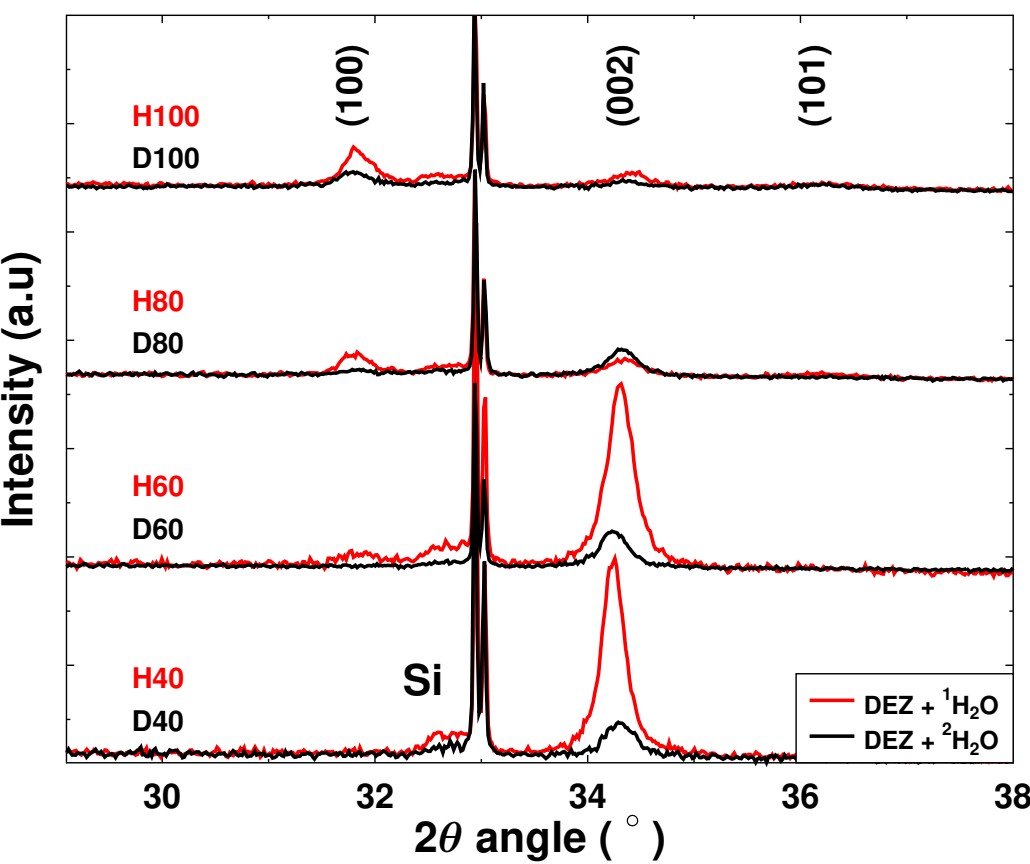

**Figure 2.** XRD patterns of selected ZnO films deposited at different temperatures with both $^1H_2O$ (red) and $^2H_2O$ (black). Films deposited with $^2H_2O$ are thinner than the films deposited with $^1H_2O$, which affects the peak intensity. The double peak at 33° originates from the silicon substrate.

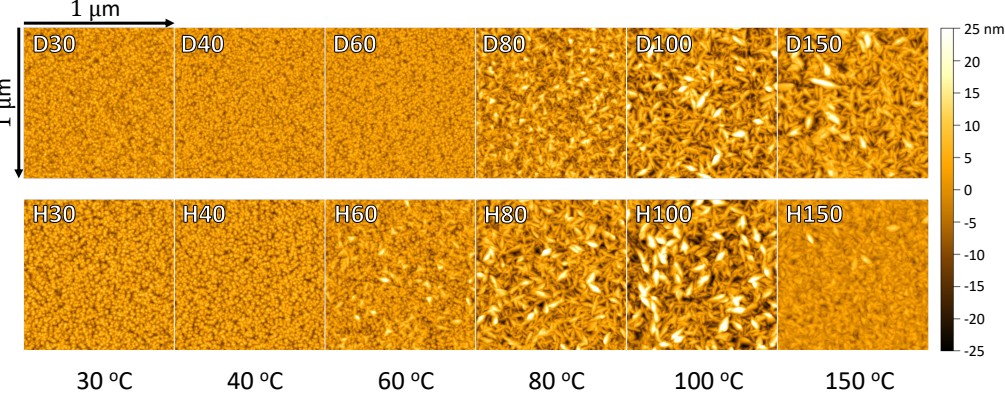

**Figure 3.** AFM micrographs of ZnO films deposited with DEZ and either $^1H_2O$ or $^2H_2O$ at different temperatures. The preferred crystal orientation changes as the temperature is raised. The lengths of the sides in each image are 1 µm.

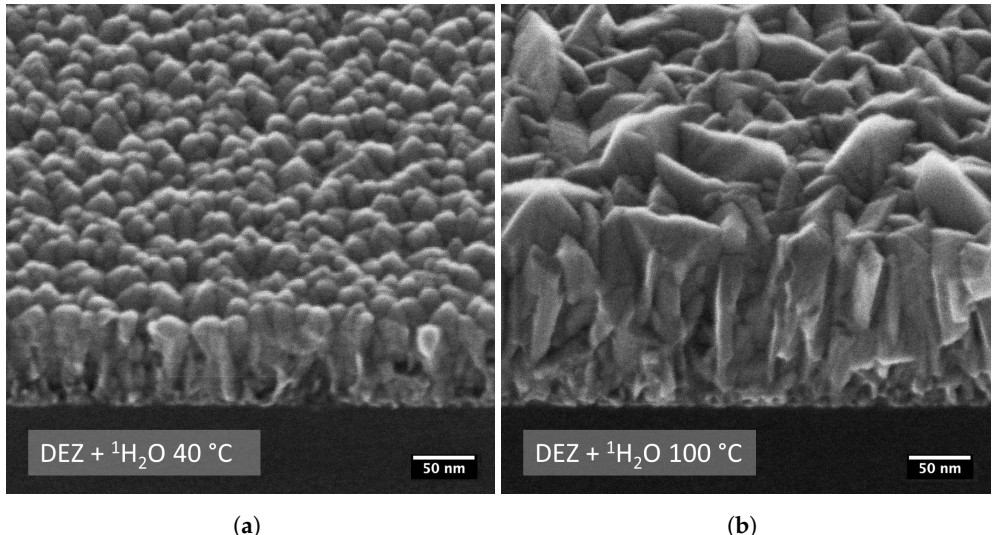

**(a)**　　　　　　　　　　　　　　　　　　　**(b)**

**Figure 4.** Cross-sectional HIM images of ZnO films deposited with normal water at (**a**) 40 °C and (**b**) 100 °C. The cleaved samples are imaged at a 45° angle relative to the beam.

Figure 5a presents the hydrogen isotope concentrations in the ZnO films as a function of the deposition temperature (also in Table 1). As expected, the total hydrogen concentration decreased with the increasing deposition temperature. The use of heavy water yielded a higher total hydrogen concentration than when using normal water, but the general trend is very similar. The higher total hydrogen concentration of the $^2H_2O$ process can be attributed to the kinetic isotope effect and similarly to the GPC and change in preferred crystal orientation; the use of $^2H_2O$ induced a temperature shift to the concentration values. A heavier hydrogen isotope raises the activation energy of the reaction, and this effect is visible especially in the lower deposition temperatures as shown in Figure 5.

In a recent work, Guziewicz et al. [36] deposited ZnO with DEZ and $^2H_2O$. Using SIMS, they found that both deuterium and hydrogen concentrations decrease as the deposition temperature is increased from 100 to 200 °C, which is in agreement with our findings. Their $^1H$ and $^2H$ concentrations are slightly lower than in our measurements. The difference is evident especially in ZnO films deposited with heavy water at 100 °C. According to our study, these films contain 1.6 at.% deuterium, while Guziewicz et al. report a value around 0.5 at.%. A possible explanation for this discrepancy could be the different pulsing and purging times used in our studies. For example, the purging time can affect the impurity composition significantly, as discussed later.

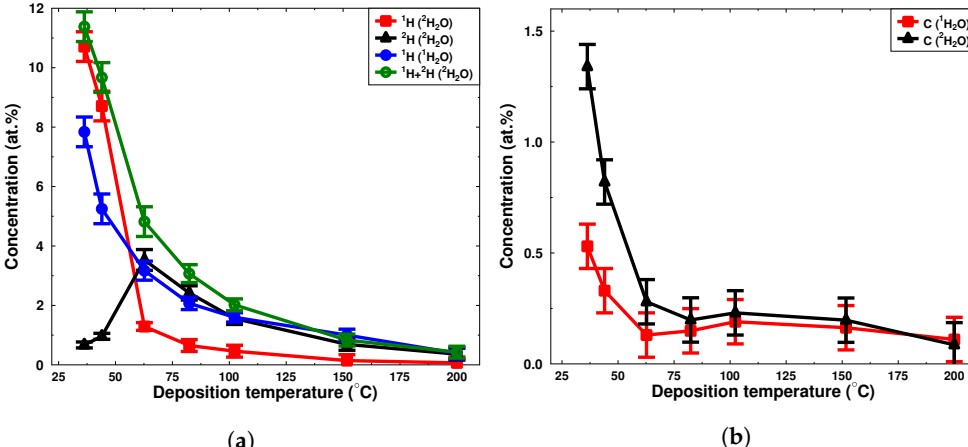

**(a)**　　　　　　　　　　　　　　　　　　　**(b)**

**Figure 5.** (**a**) Hydrogen isotope and (**b**) carbon concentration of the films deposited at different temperatures using 150 ms DEZ and 500 ms $^1H_2O$ or $^2H_2O$ pulses. Purging times after the precursor pulses were 10 and 20 s, respectively.

Guziewicz et al. [36] studied the $^1$H/$^2$H incorporation above 100 °C. Interestingly, the major hydrogen isotope changed from deuterium to hydrogen when the deposition temperature was decreased below 50 °C, as seen in Figure 5a. A similar phenomenon was observed in our previous study when Al$_2$O$_3$ films were deposited from trimethyl aluminum (TMA) and heavy water [37]. Both TMA and DEZ are organometallic alkyls whose chemical characteristics could be expected to be similar. However, the TMA + H$_2$O process produces amorphous films even at 600 °C [17], whereas ZnO films deposited using DEZ+H$_2$O are crystalline even near the room temperature, as seen in Figures 2 and 3 and shown by Malm et al. earlier [10]. Therefore, the crystallinity of the film does not seem to be a contributing factor for impurity incorporation.

The high $^1$H concentration at low temperature can be attributed to persistent ethyl groups [23,25] which behave similarly to persistent methyl groups in the case of the TMA+H$_2$O process [22,34]. In these earlier studies, it was shown both computationally and experimentally that at low temperatures water is not reactive enough to remove all the methyl/ethyl groups from the surface. The persistent alkyl-groups remain at the surface even after multiple water exposures. There is also some evidence for co-operative mechanisms of surface groups, as it has been shown that multiple methyl/ethyl groups react with a lower reaction barrier compared to isolated groups [24,50]. These isolated alkyl groups are then buried into the film during the next ALD cycles.

The carbon concentration of the ZnO films follows a similar increasing trend with the hydrogen concentration as the deposition temperature is decreased, as seen in the Figure 5b, and the carbon concentration starts to increase rapidly as the deposition temperature goes below 50 °C. However, it must be noted that the films contain much more $^1$H or much less carbon than what would be expected if these impurities only originated from persistent ethyl groups. If hydrogen originated mainly from the unreacted ethyl groups, the majority of the carbon in the film would need to escape via some yet unknown mechanism. One possible mechanism studied by Weckman and Laasonen was the $\beta$-elimination of two Zn-CH$_2$CH$_3$ surface groups following a release of gaseous C$_2$H$_4$ and C$_2$H$_6$. However, according to their calculations, the activation energy (2.52 eV) for $\beta$-elimination is too high to play any significant role on skewing the C/$^1$H ratio [25].

At low temperatures, the reduced desorption rate of molecular water from the surface is thought to be a problem for ideal ALD-growth. The growth would therefore not be self-limited, as more than a monolayer of water stays on the surface even after long purging times. However, as mentioned above, a decrease in the deposition temperature (Figure 5a) or in the purging time (Figure 6b) decreases the amount of deuterium dramatically as the concentration of hydrogen increases. It would be an easy assumption to make that the physisorbed heavy water would lead to a higher deuterium concentration and/or to uncontrolled chemical vapor deposition-like (CVD) growth. However, no evidence of uncontrolled CVD growth with very high GPC was found, as can be seen in Figure 1a,b. More interestingly, in our previous study, the TMA +$^2$H$_2$O process showed CVD-like growth already at 60 °C [37]. In addition, the Al$_2$O$_3$ films investigated in that study contained roughly two times more hydrogen than ZnO films deposited at the same temperature. It could be assumed that if the uncontrolled growth were mainly due to the adsorption of multiple layers of water, the CVD-growth should be clearly visible for both DEZ and TMA at equivalent temperature. Thus, our observations indicate that there exists either some DEZ/TMA related CVD-growth or the water sticks on the Al$_2$O$_3$ surface more tightly than on the ZnO surface.

The effect of the purging time on the impurity composition of the ZnO films was studied by pulsing DEZ and $^2$H$_2$O at 60 °C with different purging schemes. Both hydrogen and deuterium concentrations and corresponding carbon concentrations are tabulated in Table 2 and plotted in Figure 6a,b, respectively. The effect of the purging shows a very similar trend as that obtained by changing the deposition temperature. Longer purging times resulted in lower $^2$H and $^1$H concentrations, while very short purging times resulted in a dramatic drop in $^2$H and an increase in $^1$H concentrations. The higher $^1$H concentration was also correlated with the higher carbon concentration. These results indicate that the

change of the primary hydrogen isotope at low temperatures is also related to the purge time, as a shortening of the purge time has a similar effect to decreasing the deposition temperature. The effect of the purging on the roughness and crystallinity of the films was also studied with AFM (Table 2), but no noticeable difference on surface morphology was observed between the samples with different purging times.

For short purging times, the cause of the low deuterium and high hydrogen concentrations is more difficult to explain. Cai et al. used a quartz crystal microbalance (QCM) to measure the mass change during the DEZ and $H_2O$ pulses [11]. In principle, the mass change during the water pulse should be negative, since the heavier ethyl-group is replaced by a lighter OH-group. However, Cai et al. observed that the mass change during the water pulse was slightly positive. Similar results have previously been reported also by Yousfi et al. [20] and Elam and George [51]. Elam and George concluded that the DEZ molecule reacts with more than one OH-group, leaving the bare zinc atoms on the surface. Water can then react with these undercoordinated Zn atoms, balancing the mass loss from ethyl–water exchange reactions. According to Cai et al., approximately 1.5 OH-groups react with each DEZ molecule when the deposition temperature is between 80 and 250 °C, but at 30 °C, DEZ reacts with two OH-groups. This could well explain our observation of the smaller amount of deuterium at low temperatures. On the other hand, DEZ reacting with the OH-groups should also lead to low [1]H and C concentrations, which contradicts our results. In addition, Weckman and Laasonen calculated that the second ligand exchange leading to bare zinc atoms has a high reaction barrier, making the second ligand exchange feasible only at higher temperatures [24].

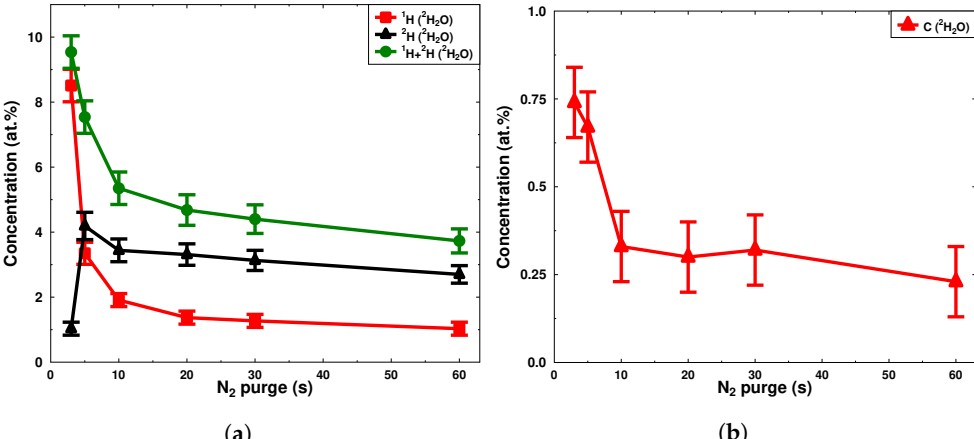

(**a**)                                                                  (**b**)

**Figure 6.** (**a**) Hydrogen and (**b**) carbon concentration as a function of the $N_2$ purge time. All the samples were deposited at 60 °C using 150 ms DEZ pulses and 500 ms $^2H_2O$ pulses. The purging time between the pulses was varied. The purging times after both precursors were kept the same when the purging time was 3, 5 or 10 s. For longer purging times, only the purge after $^2H_2O$ was increased, and the purge after DEZ was kept at 10 s.

A question arises regarding whether the purging of DEZ and $^2H_2O$ is more important than the deposition temperature for the incorporation of the impurities and $^1H/^2H$ concentrations. Therefore, more detailed purging experiments were performed at 60 °C with purging schemes of 3/3 s, 3/10 s, 10/3 s and 10/10 s after DEZ and $^2H_2O$ pulses, respectively. Results shown in Figure 7a indicate that there was a small preference for a lower total hydrogen concentration when purging after a longer DEZ pulse. However, there was no significant difference in $^1H$ or $^2H$ concentrations if the purging was changed from 3/10 s to 10/3 s. It seems that the total purging time of the ALD cycle is a more important factor in the hydrogen and deuterium incorporation. A longer purge also led to a somewhat higher GPC, as shown in Figure 7b. It is also possible that the ligands from the previous pulse, which had not yet reacted, or by-products which had not yet desorbed from the surface, could block reactive sites when the next precursor pulse arrived.

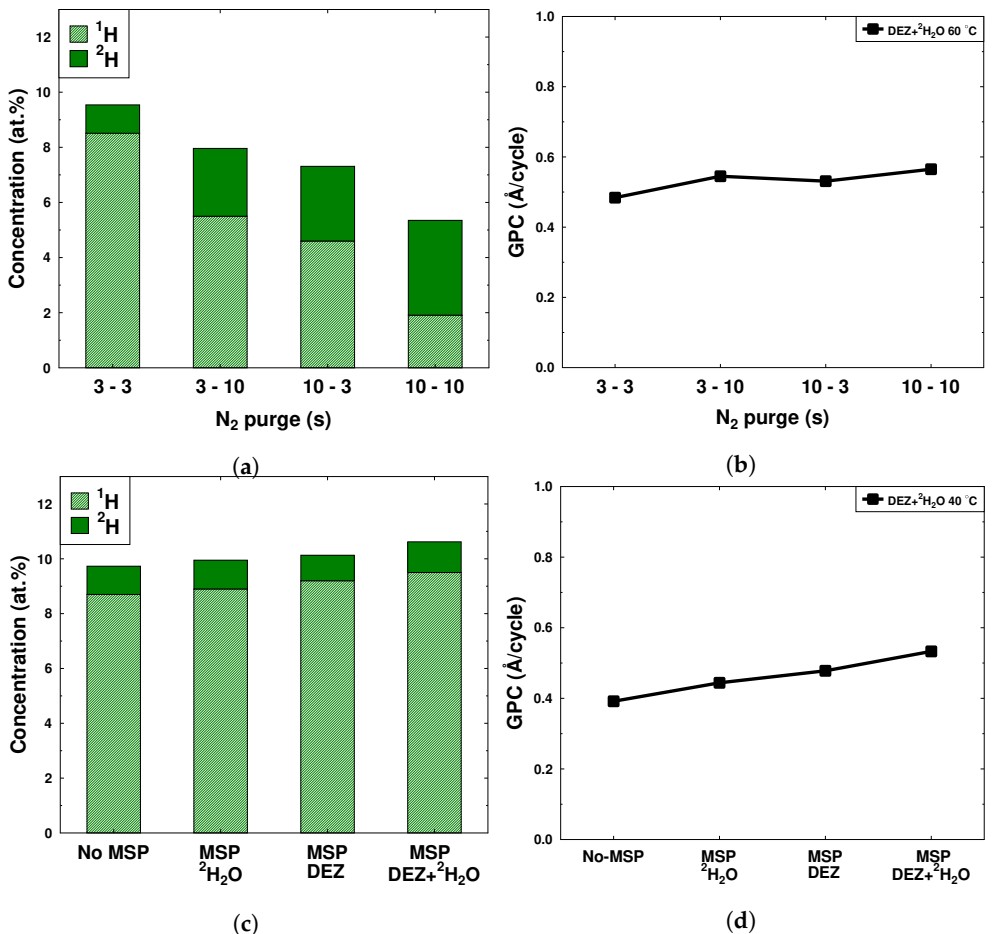

**Figure 7.** (**a**) Hydrogen and deuterium concentration with different $N_2$ purge schemes deposited at 60 °C. (**b**) Growth per cycle with different $N_2$ purging schemes deposited at 60 °C. (**c**) Hydrogen and deuterium concentration when multiple short pulses (see text) were used at 40 °C. (**d**) Growth per cycle using multiple short pulsing at 40 °C.

We also studied whether the transient steric hindrance proposed by Wang et al. [52] could explain our data. Wang et al. deposited ZnO films from DEZ and $^1H_2O$ using multiple short pulses (MSP), and they divided one pulse into shorter separated pulses while keeping the total pulse time (i.e., precursor exposure) the same. Using the MSP increased the growth per cycle, and their conclusion was that the slow desorption of byproducts and unreacted precursor molecules blocked some of the reactive sites when a single pulse was used. Multiple short pulses gives more time for desorption, allowing more precursors to adsorb, leading to a higher overall intake of material per cycle. It is known from the QCM studies that the decrease of the surface species (i.e., weight) can take seconds even at 177 °C [51].

In order to study this hypothesis, ZnO films were deposited at 40 °C using a recipe in which the pulses were divided into three shorter pulses so that the total precursor dose remained unchanged. This was done using $3 \times 50$ ms pulses for DEZ and $3 \times 167$ ms for heavy water. The purge time between the short pulses was set to 1 s. In order to keep the total cycle time the same as in the previous depositions, purging times of 8 s and 18 s were used instead of 10 s and 20 s after DEZ and heavy water, respectively. Three different samples were deposited: MSP for both DEZ and water, MSP only with $^2H_2O$ and MSP only with DEZ.

Hydrogen and deuterium concentrations of the films deposited with the MSP process along with conventional single pulse deposition are shown in Figure 7c. Multiple short pulses did not significantly change the $^2H$ concentration but slightly increased the $^1H$ concentration in each case. The highest total hydrogen concentration was measured when MSP was used for both DEZ and $^2H_2O$, and at the same time, the GPC increased more

than 25% from 0.40 to 0.53 Å/cycle. These two results are somewhat contradictory, as an increase in the GPC would be an indication of the transient steric hindrance proposed by Wang et al. [52]. On the other hand, this should lead to a film with fewer impurities, but our results indicated the opposite. The reason behind the contradiction remains unknown and requires further studies. It seems that the total purging time is the most significant factor affecting the film composition, as shown in Figure 7a. Nevertheless, the MSP seems to increase the GPC significantly without increasing the cycle time, and the faster low-temperature deposition is beneficial for industrial applications.

In order to study the hydrogen (deuterium) migration and exchange reactions in the film, samples deposited with $^2H_2O$ were measured again after two months of storage in ambient conditions. The results (Figure 8) show that the sum of the hydrogen and deuterium in the film increased slightly during the storage. Simultaneously, the $^2H$ concentration in the film decreased considerably while the $^1H$ concentration increased. Especially in the sample deposited at 60 °C, which had the highest deuterium concentration to begin with, almost all the deuterium was replaced with hydrogen.

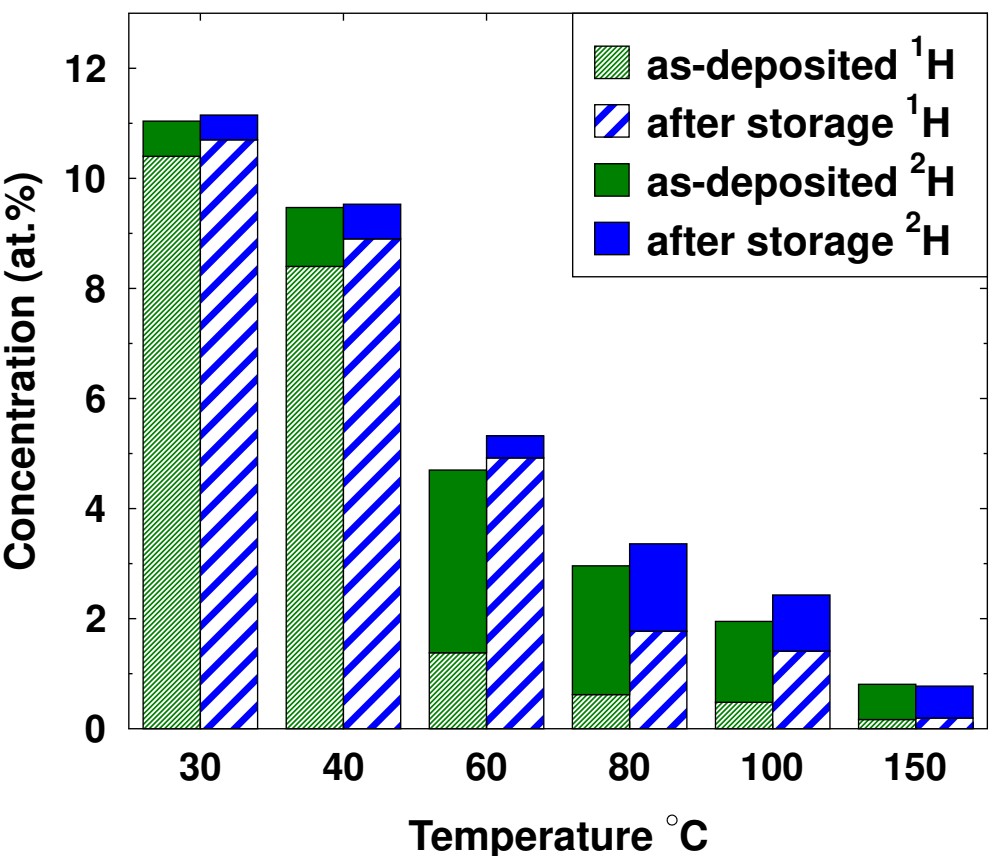

**Figure 8.** Change in the $^1H/^2H$ concentration after two months of storage at ambient conditions. For example, in the far left bar, the film deposited at 30 °C contains 10.4 at.% $^1H$ and 0.64 at.% $^2H$ immediately after the deposition and 10.7 at.% $^1H$ and 0.45 at.% $^2H$ after the storage.

This decrease in the deuterium concentration was most likely caused by an exchange reaction between the deuterium atoms in the film surface and hydrogen atoms from either molecular hydrogen $H_2$ or from water vapor, which are found in air. Since these gases contain only a very minor amount of deuterium, the deuterium concentration in the film would decrease. This exchange does not only take place in the film surface, but the deuterium concentration also decreases through the cross-section of the film, even if it is 100 nm thick. Analogous results were obtained in our previous study with $Al_2O_3$ [37]. However, the ALD ZnO films are crystalline whereas $Al_2O_3$ films are amorphous.

## 4. Conclusions

The atomic layer deposition of ZnO at low temperatures is a complex process. Our study shows that there are probably multiple processes that occur at the same time, including the fast primary reactions of the ALD growth and the slower secondary reactions that can take place even below the outermost layer of atoms. This is demonstrated by varying the purging time, which has a significant effect on the impurity levels of the films. All the films deposited in this study had a uniform thickness throughout the reactor and did not show any hint of CVD-growth even though the elemental compositions of the films were significantly different. This can influence the film properties. Therefore, it is important to describe the deposition conditions in great detail. This also means that great care must be taken when the results from multiple studies are compared, especially at low deposition temperatures.

The use of stable isotopes gives valuable information regarding the reactions, especially when the deposition conditions are not ideal; i.e., at low temperatures where the reaction rate can be slow and adequate purging times impractically long. The high $^1$H and low $^2$H concentrations in ZnO films at low temperatures and with short purging times indicate the existence of some unidentified mechanism that may govern the growth in these non-ideal conditions. The second ligand exchange reaction suggested by Cai et al. [11] was found to be an unlikely candidate at low temperatures, since $^1$H and C concentrations increased rapidly at low temperature.

The high $^1$H and low $^2$H concentration at low deposition temperatures could be an indication of persistent non-reactive ligands, as suggested by Weckman et al. [25] and Mackus et al. [23]. More reactive DEZ would remove practically all the possible OH-groups, leading to decreased $^2$H and increased $^1$H concentrations. However, there are two observations contradicting this explanation: firstly, there was a lower concentration of carbon in the film than what would be expected based on the $^1$H concentration or vice versa; secondly, shortening the purging time led to a similar effect as decreasing the deposition temperature, which also implies the existence of some slower yet unknown reaction mechanism, as discussed above.

There is also a transient steric hindrance component which decreases the GPC at low temperature. Multiple short pulses can be used to tackle this problem, as a higher GPC can be achieved without increasing the ALD cycle time, making the low-temperature process faster. Changing from a single pulse to the MSP does not, however, change the impurity concentrations of the films. This can be achieved only through longer purging, as shown above. The longer purging times also lead to a somewhat higher GPC. Therefore, there might be a transient steric hindrance component related also to this slower process of removing the persistent methyl groups from the film. These species can also block the reactive sites which become available to the precursor only after the long purging.

The kinetic isotope effect due to heavier deuterium induces an effect similar to decreasing the deposition temperature, which must be taken into account when comparing our results with results from different studies. However, replacing normal water with heavy water seems to reproduce similar trends in terms of the GPC, impurity concentration and crystallinity, validating the use of heavy water when studying the DEZ + $H_2O$ ALD-process.

**Supplementary Materials:** The following are available online at https://www.mdpi.com/2079-641 2/11/5/542/s1, Figure S1: HIM-$^1H_2O$, Figure S2: HIM-$^2H_2O$.

**Author Contributions:** Investigation, S.K., M.L. and K.A.; Writing—original draft, S.K.; Writing—review and editing M.L., K.A. and T.S. All authors have read and agreed to the published version of the manuscript.

**Funding:** This research received no external funding.

**Institutional Review Board Statement:** Not applicable.

**Informed Consent Statement:** Not applicable.

**Data Availability Statement:** Not applicable.

**Conflicts of Interest:** The authors declare no conflict of interest.

## Abbreviations

The following abbreviations are used in this manuscript:

| | |
|---|---|
| AFM | Atomic force microscopy |
| ALD | Atomic layer deposition |
| CVD | Chemical vapor deposition |
| DEZ | Diethylzinc |
| GPC | Growth per cycle |
| HIM | Helium ion microscopy |
| IR | Infrared |
| MSP | Multiple short pulses |
| OLED | Organic light emitting diode |
| QCM | Quartz crystal microbalance |
| QMS | Quadrupole mass spectrometry |
| RMS | Root-mean-square |
| SIMS | Secondary ion mass spectrometry |
| TMA | Trimethylaluminium |
| ToF-ERDA | Time-of-flight elastic recoil detection analysis |
| XRD | X-ray diffraction |

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
