# Peer review of "Hydrogen and Deuterium Incorporation in ZnO Films Grown by Atomic Layer Deposition"

_coatings, doi:10.3390/coatings11050542_

Round 1
Reviewer 1 Report
The manuscript is interesting and has potential to attract more attention in the ZnO research community.The manuscript can be accepted in its present form.
The following comments can be considered.
- Please modify the introduction with the related research already presented and the importance in the ZnO with deposition and morphology.
- A table may be used to show the importance of ZnO for the mention applications as compared to other oxides.
Author Response
The introduction now contains more ALD specific ZnO examples and references. The unique assets of ALD are emphasized in order to justify the subject of the study but it was considered to be more practical to use still text without extra table.
Reviewer 2 Report
The authors have provided the revised version of the manuscript. Although, the authors did not provide satisfactory answer to some of the raised queries. However, looking at the overall quality of the manuscript shows that it still has potential. Hence, I would like to accept the paper publication after minor revision of the following comments.
1- The manuscript, especially the revised paragraphs and added text, has text errors, typos and structural mistakes. Moderate English changes required before publication. Also, always provide the proper line numbers to the reviewers to address the comments. Good luck.
Author Response
The manuscript has now been spell checked and all the found typos and errors have been addressed to the best of our knowledge.
Reviewer 3 Report
Manuscript Number: 1181155
Title: Hydrogen and deuterium incorporation in ZnO films grown by atomic layer deposition
Article Type: Research article
Recommendation: Major Revision
Comments to authors: The authors presented an interesting work about atomic layer deposition (ALD) of ZnO. The work illustrates the impact of different parameters like temperature and purging time on ZnO ALD. The results are well supported by characterization and references. The work provides great insights and can be published after making the following changes.
- On page 3 line 98, author states “All the deposited ZnO films were found to be slightly oxygen rich”. Authors should explain the reason behind this.
- On page 3 line 120, author states “Increasing the purging time slightly increases the growth per cycle.” However as per the literature cited (reference 36), there was a decrease in the GPC with purging time. Though the temperatures are considerably different, still author need to explain better why the trends are different.
- On page 3 author states, at 60 0C, the loss of OH- groups does not play a significant role. Overall it appears author is trying to say that the decrease in GPC with purging time from reference is due to high temperature operation and at low temperature (used in the manuscript) the thickness increased. It seems partially correct but at low temperature the precursor will also react slowly and can lead to smaller thickness. Then how did author claim an increase in thickness with purging time?
- On page 5 line 145-147, author states that “at low temperatures (40 â—¦C and 60 â—¦ C) preferred orientation of the crystals is (002) that transforms to (100) at higher temperature (100 â—¦ C). Author have explained it well but authors are advised to show XRD results first and then AFM results because it is easier to understand about crystallinity from XRD and then it can facilitate readers to understand their explanation of AFM better.
- On page 9 line 283-285 author states “Multiple short pulses did not significantly change the 2H concentration but slightly increased the 1H concentration in each case”. Authors are advised to discuss this in more details.
Author Response
1. The following text to explain the excess oxygen was added:
However, oxygen rich ZnO films deposited with ALD, especially at low deposition temperatures, have been reported before [10,27,43–45]. High oxygen concentration in the ZnO films have been attributed to zinc vacancies [44] in the crystal as well as for oxygen interstitials [43]. In addition, impurity hydrogen has been proposed to occupy Zn-sites in the crystal [27]. Article now contains references which report oxygen-rich ZnO films and the possible origin of the excess oxygen or missing zinc is now properly addressed.
2. and 3. These two are excellent comments. The increase of GPC as a function of purging time is now more thoroughly discussed. We also note that the trend of increasing GPC is opposite to what one might expect for low temperature ALD. While the definite cause of the phenomena remains unknown, the slow desorption of by-products is given as plausible explanation.
4. The order of AFM and XRD figures is interchanged and the text is edited accordingly.
5. The authors cannot give a fully satisfactory explanation why the amount of 1H increases when MSP is used. The change in concentration is small, but most probably a real one and it is larger than the statistical error of our measurements. A sentence stating our lack of knowledge on this matter is added to the article.
Reviewer 4 Report
This work studies Zinc oxide (ZnO) thin films grown by atomic layer deposition where normal water or heavy water (2H2O) have been used as oxidants. Useful information about reaction mechanisms and hydrogen incorporation in dependence on deposition temperature (from 30 to 200 â—¦C) have been obtained. The paper is comprehensibly written and is scientifically meaningful. The paper will benefit if the following issues are also addressed:
- What are the advantages of using heavy water?
- What are the anticipated improvements of film properties when using heavy water?
- How some basic properties (e.g. transparency, bandgap, etc.) change when heavy water is used?
Author Response
The purpose of using heavy water, instead of normal water, was not to demonstrate improvements or changes in film properties but rather to study the hydrogen incorporation from two hydrogen containing precursors at different deposition conditions. The film quality in terms of impurity hydrogen concentration and crystallinity was actually worse with the heavy water. The comparison between the two waters was made in order to validate the use of heavy water. The two waters are chemically similar, but as demonstrated in the article, the kinetic isotope effect produces a “temperature shift” to the film composition. This needs to be carefully considered when comparing the results between the processes with two waters and drawing conclusions of the heavy water depositions.
Round 2
Reviewer 2 Report
The authors present hydrogen and deuterium incorporation in ZnO films grown by atomic layer deposition methods. The films were grown with normal water and heavy water. The authors studied the hydrogen concentration in the films at different growth temperatures. The ides in interesting; however, the technical aspects of the manuscript need further explanation and experimentation. Hence, I would like to reconsider the manuscript based on the major revision of the following comments.
1- Introduction: The introductory paragraph about ALD is very generic. It would be better if the authors provide a comparative analysis of different ZnO growth techniques and conclude with the reason why they preferred ALD upon other methods of ZnO growth.
2- The information in line 25-34 is also very generic. It is better to cover the information via a bibliographic reference.
3- Where did the authors get the information in line 98 to 101? Specify the source in the manuscript. Secondly, the ZnO samples are normally oxygen deficient, which is the basic growth habit of ZnO. ZnO is naturally n-type only because of the reason that it has sufficient oxygen donor vacancies. Furthermore, the crystal structure is fraught with plethora of Zn interstitials. On the contrary, the authors are claiming the opposite. How would the authors make such a gigantic claim of oxygen rich ZnO?
4- What is the specific morphology grown by the authors in the claimed growth conditions. Support your answer by providing the SEM images. Also provide the cross-sectional SEM images to confirm the change in orientation from 002 to 100. The AFM results are neither suffice not clear to establish the claim.
5- Figure 3: The two large peaks at 33 degrees are unusual to ZnO. Define the nature of these peaks. Also, provide the JCPDS card number for peak comparison.
6- It is very important to study the crystal growth habit and defects chemistry of the crystals grown with two waters. Hence, I would recommend the authors to characterize the samples with transmission electron microscopy and photoluminescence spectroscopy for clarity.
Author Response
- The topic of this paper was to understand better the ALD growth of ZnO, hence the focus to ALD ZnO examples. The introduction now contains more ALD specific ZnO applications as examples. However, the basis of this study was to demonstrate hydrogen incorporation in different ALD conditions. ALD is well established method in thin film deposition, especially in integrated circuit applications. Zinc oxide deposited with DEZ and water is one of the most studied ALD material and we feel that studying and reporting the impurity incorporation in different deposition conditions is valuable.
- The ideal reaction mechanism could be presented as a reference, but we find it beneficial to remind readers of the possible sources of impurity hydrogen. In addition, it shows that in a perfect world films should not contain any hydrogen while in real life there can be up to 10 at. % of hydrogen in the film. After consideration, for the sake of clarity we decided still to keep the reaction mechanisms in the paper.
- This is a fair comment. As pointed out in the text, the ion beam analysis technique TOF-ERDA we have used to study the composition in this study slightly underestimates the Zn content due to multiple and plural scattering (see https://doi.org/10.1016/S0168-583X(00)00435-3 ). Because of this, the O/Zn may be a little overestimated. However, oxygen rich ALD ZnO films have been reported before, and this seems to be the case especially at low deposition temperatures. Manuscript now contains citations to articles reporting oxygen rich ZnO films deposited with ALD.
- We agree with the reviewer that the AFM results are not sufficient to make such a claim. However, we believe that the combination of AFM and powder XRD is sufficient to confirm temperature dependent change in preferential orientation reported in several papers before us. In addition, we did now helium ion microscopy and provided new images of selected films in order to confirm the orientation change.
- The large peaks at 33 degrees originate from silicon substrate as mentioned in the caption. This is now marked also in the figure in order to reduce confusion.
- Within this 10 day schedule we were unable to perform TEM or PL spectroscopy measurements and we fully agree that from applications point of view these are of course important characterization methods. However, this article tried to concentrate more on the fundamentals of ALD grown films and impurity incorporation. The crystal growth habit and defects chemistry would be excellent subjects for further study.
Reviewer 3 Report
The authors have addressed all the comments properly and the manuscript can be published now.
Author Response
The authors want to thank the reviewer for the constructive comments.